# Learning to Influence Human Behavior
# with Offline Reinforcement Learning

**Joey Hong**    **Sergey Levine**    **Anca Dragan**
UC Berkeley
{joey_hong,sergey.levine,anca}@berkeley.edu

## Abstract

When interacting with people, AI agents do not just influence the state of the world – they also influence the actions people take in response to the agent, and even their underlying intentions and strategies. Accounting for and leveraging this influence has mostly been studied in settings where it is sufficient to assume that human behavior is *near-optimal*: competitive games, or general-sum settings like autonomous driving alongside human drivers. Instead, we focus on influence in settings where there is a need to capture human *suboptimality*. For instance, imagine a collaborative task in which, due either to cognitive biases or lack of information, people do not perform very well – how could an agent influence them towards more optimal behavior? Assuming near-optimal human behavior will not work here, and so the agent needs to learn from real human data. But experimenting online with humans is potentially unsafe, and creating a high-fidelity simulator of the environment is often impractical. Hence, we focus on learning from an *offline* dataset of human-human interactions. Our observation is that *offline reinforcement learning* (RL) can learn to effectively influence suboptimal humans by extending and combining elements of observed human-human behavior. We demonstrate that offline RL can solve two challenges with effective influence. First, we show that by learning from a dataset of suboptimal human-human interaction on a variety of tasks – none of which contains examples of successful influence – an agent can learn influence strategies to steer humans towards better performance *even on new tasks*. Second, we show that by also modeling and conditioning on human behavior, offline RL can learn to affect not just the human's actions but also their underlying strategy, and adapt to changes in their strategy.

## 1   Introduction

In many AI applications, including games [27], healthcare [26, 32], recommender systems [1], and robotics [15], agents interact with people and end up *influencing* human behavior. Accounting for and leveraging this influence has mainly been studied in settings where it is sufficient to model human behavior as near-optimal [13, 8], like in games such as Go [27], or in autonomous driving settings where agents try to influence people to slow down and make space [25].

In contrast, we target settings where influence is important but real humans might not behave in strategically optimal or even rational ways, such as cooperative scenarios with other non-expert humans, or social scenarios that are not inherently strategic, like dialogue. For instance, imagine a human and robot collaborating to cook a meal. The human might start chopping the tomatoes to make a salad because there are tomatoes conveniently nearby, but it might be much more efficient to let the robot make the salad while the human plates the main course (because the robot is less adept at plating). To try to get the human to plate, the robot might put a plate right next to them to make the plating task more appealing, or it might even block the human from accessing the tomatoes. Such

37th Conference on Neural Information Processing Systems (NeurIPS 2023).

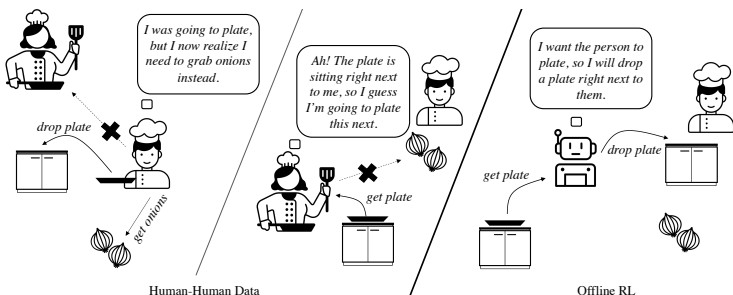

Figure 1: Informative example of how an agent may deduce influencing strategies from offline data.

strategies would not be needed with optimally rational humans who understand precisely what they should do, and therefore they must account for the statistical properties of real suboptimal human behavior that allow humans to be influenced by the robot's behavior.

Prior work in influencing suboptimal behavior in humans has thus far approximated human behavior with scripted or heuristic policies [34, 16]. However, humans have a myriad of cognitive biases [30, 10] that can make their behavior sub-optimal. This makes hand-designing a simulator for how humans behave very difficult, as one cannot write heuristics for all these possible biases. In turn, it implies that learning influence will require relying on actual human data. But interacting with people online to figure out what works and what does not is unsafe or unrealistic in many domains.So how can we attain effective influence when given only a data set of (suboptimal) *human-human* interaction?

Our key observation is that an agent can learn influencing policies even when these are not demonstrated in the data. In particular, *offline reinforcement learning* (RL) can learn to influence by "stitching" or combining different components of human behavior seen in a diverse human-human interaction dataset. A simple but illustrative example that illustrates this in the collaborative cooking setting is in Figure 1: in some human-human interaction, a human makes a mistake and realizes that they want to go grab onions instead of plating, and drops the plate on the counter; in a (potentially different) interaction, a human may pick up the plate that is conveniently placed right next to them, abandoning their current next goal in favor of plating. By extending those two behaviors, an agent can glean that placing a plate next to the human might influence them towards plating. As an added benefit, this can happen without access to a simulator or model of the physical world.

The main contributions of our work is to demonstrate how offline RL can address two important challenges when it comes to influencing suboptimal human behavior. First, we show that existing offline RL methods can deduce strategies to influence a human's actions despite not seeing any such strategies in the training data, either because the humans were doing different tasks, did not attempt influence, or did not think of a successful influence strategy. Second, by augmenting offline RL algorithms with an estimate of the human's *latent strategy*, we can enable the RL agent to go beyond influencing human actions only, to influencing their actual strategy. For example, the RL agent might figure out that if it repeatedly blocks a person from reaching a particular ingredient, that person will change their strategy to avoid that ingredient in the future, even when it is no longer blocked, thus altering their strategy for the rest of the episode.Our work can be seen as expanding the potential of offline RL by showing how it can be leveraged in human interaction, trained from human-human data, and conducive to good performance when deployed alongside real humans.

## 2 Related Work

**Multi-agent RL.** RL has been applied extensively to multi-agent settings, where multiple agents take actions in a competitive or cooperative game [31, 24, 14, 36]. A standard technique in MARL involves modeling the joint effect of the all agents' actions in the environment [11]. Many such methods utilize centralized training, which accounts for the actions of other agents through a centralized critic [24, 12]. Other related approaches add explicit communication channels between agents so that agents can share their policy parameters or gradient updates [7, 23]. These methods are typically concerned with jointly training a collection of autonomous agents. In contrast, our work focuses

on training agents that interact with humans, who are not under our direct control (and thus cannot access their policies), and often behave suboptimally or irrationally.

**RL with humans.** In recent years, increased attention has been directed toward designing agents that interact with humans. Early works in this setting train agents that play against humans in competitive games, such as Go [3], Poker [5], and Diplomacy [4], and aim to exceed the skill level of their human opponents. Such methods typically rely on computing best responses against a (near-) optimal agent. However, in many realistic tasks, modeling the human as near-optimal is insufficient, such as in collaborative or social settings [6]. We focus on tasks where modeling and improving suboptimal human behavior is important for success. Related to our work, Carroll et al. [6] investigated coordination with humans in the *Overcooked* environment. However, we deviate from this in several substantial ways. First, we focus on influence to improve suboptimal human behavior rather than simply accounting for it; we do this by considering tasks where an agent must make its human partner behave differently in order to succeed. Second, we do not assume access to an environment simulator, and must learn completely from prior human-human interactions.

**Multi-agent influence.** We investigate whether we can train agents that learn to influence humans to behave more optimally. Prior works have also considered using influence to improve competition or cooperation, and have proposed both model-free and model-based approaches. Model-free approaches anticipate how other agents update their policy without explicitly learning a model of the dynamics. Notably, in competitive games, LOLA [8] and its more recent improvements [9, 21] estimate the opponent's policy updates, and use it to inform the policy updates of the agent in a recursive manner. Model-based approaches, on the other hand, consider a dynamics model of how other agent's policies change. Existing works such as LILI do so practically by learning a low-level representation of other agents' behavior, and condition the learned agent's behavior on these representations [16, 33]. However, LILI is (1) trained and evaluated on scripted and heuristic policies, which display substantially less complex behavior than real humans, (2) require online interactions, and (3) assume behavior changes only between episodes. In this paper, we address all these limitations by proposing offline RL solely on prior human-human interactions, but can also learn agents that adapt its influence even when human behavior changes within episodes.

## 3 Preliminaries

The goal in RL is to learn a policy $\pi(\cdot|\mathbf{s})$ that maximizes the expected cumulative discounted reward in a Markov decision process (MDP), which is defined by a tuple $(\mathcal{S}, \mathcal{A}, P, R, \gamma, H)$. $\mathcal{S}, \mathcal{A}$ represent state and action spaces, $P(\mathbf{s}'|\mathbf{s}, \mathbf{a})$ and $R(\mathbf{s}, \mathbf{a})$ represent the dynamics and reward distribution, $\gamma \in (0, 1)$ represents the discount factor, and $H$ (optionally) is the horizon of the episode.

**RL with hidden information.** In our problem formulation, we are concerned with tasks that require interacting another agent, such as a person, whose policy and behavior is unknown. We model the other agent as having some low-dimensional *latent strategy*, $\mathbf{z} \in \mathcal{Z}$, which determines their current behavior (e.g., goal, plan, etc.). We formulate this problem as a special case of a partially observable MDP (POMDP), which we refer to as a *hidden MDP* (Hi-MDP). A Hi-MDP is defined by $(\mathcal{S}, \mathcal{A}, \mathcal{Z}, P, P^z, R, \gamma, H)$, where $\mathcal{Z}$ is the space of internal states that represent the other agent's behavior, $P(\mathbf{s}'|\mathbf{s}, \mathbf{a}, \mathbf{z})$ additionally depends on the latent behavior, and $P^z(\mathbf{z}'|\mathbf{s}, \mathbf{a}, \mathbf{z})$ represents the dynamics. We refer to the agent controlled by the learned policy in the Hi-MDP as the *ego agent*. A similar MDP is studied in Xie et al. [34], but consider a $\mathbf{z}$ that is fixed within a single episode.

**Offline RL** Our approach employs offline RL to learn coordination strategies from data, without requiring human interaction or simulation. In offline RL, we are given access to a static dataset consisting of tuples $\mathcal{D} = \{(\mathbf{s}, \mathbf{a}, r, \mathbf{s}')\}$. Standard model-free offline RL algorithms are often actor-critic algorithms, which attempt to learn a Q-function $Q^\pi : \mathcal{S} \times \mathcal{A} \to \mathbb{R}$ of policy $\pi$ at all state-action pairs $(\mathbf{s}, \mathbf{a}) \in \mathcal{S} \times \mathcal{A}$, Specifically, for a policy $\pi$, its Q-value $Q^\pi(\mathbf{s}, \mathbf{a}) = \mathbb{E}_\pi \left[ \sum_{t=0}^\infty \gamma^t r_t \right]$ is its expected mean return starting from that state and action. The Q-function is the unique fixed point of the Bellman operator $\mathcal{B}^\pi$ given by: $\mathcal{B}^\pi Q(\mathbf{s}, \mathbf{a}) = r(\mathbf{s}, \mathbf{a}) + \gamma \mathbb{E}_{\mathbf{s}' \sim P(\mathbf{s}'|\mathbf{s}, \mathbf{a}), \mathbf{a}' \sim \pi(\mathbf{a}'|\mathbf{s}')} [Q(\mathbf{s}', \mathbf{a}')]$ , meaning $Q^\pi = \mathcal{B}^\pi Q^\pi$. In the offline setting, where we are limited to interactions that appear in the dataset $\mathcal{D}$, we do not have access to the true Bellman operator $\mathcal{B}^\pi$. Instead, we use the empirical Bellman operator that backs up a single sample $(\mathbf{s}, \mathbf{a}, r, \mathbf{s}') \in \mathcal{D}$, denoted as $\hat{\mathcal{B}}^\pi$. In an actor-critic algorithm, a separate policy $\pi$ is trained to maximize the Q-value of its chosen actions. To do this, actor-critic methods switch between updating a policy-dependent Q-function

$Q^\pi$ via: $\hat{Q}^{k+1} = \arg\min_Q \mathbb{E}_{\mathbf{s},\mathbf{a}\sim\mathcal{D}}\left[\left(Q(\mathbf{s},\mathbf{a}) - \hat{\mathcal{B}}^{\hat{\pi}_k}\hat{Q}^k(\mathbf{s},\mathbf{a})\right)^2\right]$ and improving the policy $\pi(\mathbf{a}|\mathbf{s})$
by solving for: $\hat{\pi}^{k+1} = \arg\max_\pi \mathbb{E}_{\mathbf{s}\sim\mathcal{D},\mathbf{a}\sim\pi_k(\mathbf{a}|\mathbf{s})}\left[\hat{Q}^{k+1}(\mathbf{s},\mathbf{a})\right]$. Standard RL algorithms such as
Q-learning suffers from distributional shift [17, 22] because the actions in $\mathcal{D}$ are derived from some
likely suboptimal policy. In our paper, we use conservative Q-learning (CQL) [18], which additionally
penalizes Q-values on out-of-distribution actions when learning the Q-function:

$$\hat{Q}^{k+1} = \arg\min_Q \ \mathbb{E}_{\mathbf{s},\mathbf{a}\sim\mathcal{D}}\left[\left(Q(\mathbf{s},\mathbf{a}) - \hat{\mathcal{B}}^{\hat{\pi}_k}\hat{Q}^k(\mathbf{s},\mathbf{a})\right)^2\right] + \alpha\left(\mathbb{E}_{\mathbf{s}\sim\mathcal{D}}\left[\log\sum_\mathbf{a}\exp(Q(\mathbf{s},\mathbf{a}))\right] - \mathbb{E}_{\mathbf{s},\mathbf{a}\sim\mathcal{D}}\left[Q(\mathbf{s},\mathbf{a})\right]\right) \ .$$

In this work, we use CQL to accomplish cooperative tasks that involve influencing other agents in
the environment. However, our work is not specific to CQL or any particular offline RL method, and
could well be adapted to other effective offline RL methods.

## 4 Human Influence in the *Overcooked* Environment

We hypothesize that offline RL can learn how
to influence suboptimal human behavior, using
solely a dataset of human-human interactions.
We test this hypothesis on an environment that
is a simplified version of the game *Overcooked*,
originally proposed in Carroll et al. [6]. We
choose this environment due to its popularity
in human-aware RL, and because humans are
known to behave suboptimally due to miscoor-
dination with their partner.

In this environment, two players are placed into
an environment consisting of 2D tiles, with ob-
jects such as stoves, ingredients (tomatoes or
onions), and obstacles, and tasked with deliv-
ering as many cooked dishes composed of ei-
ther tomatoes or onions as possible within an
episode. This involves a series of sequential
high-level actions to which both players can con-
tribute: collecting ingredients, depositing them
into cooking pots, letting the ingredients cook
into soup, collecting a plate, getting the soup on
the plate, and delivering it.

Figure 2: Illustration of the *Overcooked* environ-
ment. The goal is to place three ingredients (toma-
toes or onions) in a pot, plate the soup from the pot,
and deliver it, as many times as possible within the
time limit. The ego agent is the one in the blue hat.

Each player observes an egocentric view of the
world that includes the layout of the kitchen and
location of the other player, and at every step can perform one of six actions: `stand still`, `move`
{`up`, `down`, `left`, `right`}, `interact`. The effect of `interact` varies based on the cell which the
player is facing. To effectively complete the task, players must navigate the kitchen and interact
with objects in the correct order, all while coordinating with what their partner is doing. While the
dynamics of the environment are relatively simple, the introduction of real humans as partners allows
for complex and diverse influence strategies that cannot be found in prior works that use scripted or
heuristic policies [34, 16, 33].

In this work, we consider two complementary challenges of learning to influence suboptimal human
behavior in this environment and propose how offline RL can address them. We describe and motivate
both challenges below, and defer details of how we assess whether offline RL succeeds at each
challenge in Section 5 and 6, respectively.

**Challenge 1: Deducing New Influence Strategies.** While it is relatively straightforward to collect
human-human interactions in the *Overcooked* environment, it is comparatively hard to observe
effective influence strategies. This is because in this environment, humans are passive and will often
only respond to what their partner is doing, *e.g.*, by completing the next available task, and not try to
develop more effective coordination by influencing their partner's behavior. For instance, imagine
that the agent knows its human partner is much better suited to plating and delivering soups, *e.g.*,

due to being closer to the delivery station. An intelligent agent should try to elicit this behavior, potentially by trying to "pass" the plate to its partner. However, because in this setting, humans do not tend to intentionally try to change what their partner is doing, this specific interaction may not appear in the data. This means that influence strategies must be deduced, by extending and generalizing the suboptimal human behaviors found in the data.

We hypothesize that naive offline RL algorithms can learn new influence strategies, even using a dataset of diverse suboptimal behavior that *contains no evidence of influence*, by combining, or "stitching", components of human behaviors to reason about unseen strategies. This is done by identifying new ways to reach intermediate states that will result in successfully changing their partner's behavior, by propagating the reward signal from trajectories that contain the desired behavior through the Q-function for those states. Recall the example of influencing a human to plate a soup. The dataset may not contain trajectories where plate is passed between humans, but instead contains ones where a human puts a plate on the counter, and (potentially different) ones where the human partner picks up a nearby plate. Then, offline RL can learn that moving a plate to a counter near the human may influence them to pick it up and plate the soup.

**Challenge 2: Long-term Influence of Latent Strategies.** When influencing humans, it is often not sufficient to simply learn an influence strategy and apply it unconditionally on the human partner. Rather than getting the human to execute particular actions, it can be more desirable to achieve long-term influence by changing their underlying policy, or *latent strategy*. This is also true in the *Overcooked* environment. For example, consider the case where the agent wants the human to always transfer tomatoes to the pot rather than onions. By explicitly blocking the human from reaching the onions for an extended period of time, the human may adapt its policy and decide to only pick up tomatoes in the future. Furthermore, the agent should recognize that the human's policy has now changed, and deem that blocking the onions any further is a waste of resources. This means that effective influence requires knowing how to achieve long-term sway on the human's latent strategy.

It is evident that naive offline RL cannot learn adaptive policies. But via a natural and simple modification, *i.e.*, modeling the human's latent strategy, and training policies conditioned on that strategy, we hypothesize that offline RL can train agents that can influence and adapt to changes in the human's latent strategy. We believe this can be done even on data where humans overwhelmingly fail to coordinate successfully, likely due to incorrectly predicting their partner's behavior. This is because offline RL can still use trajectories of failed coordination to identify the different behaviors that are present and how to optimally respond to them.

## 5 Learning Influence Strategies from Diverse Behaviors

Our goal in this section is to demonstrate that an agent trained using offline RL can deduce new strategies to influence and improve human behavior, by smartly recombining parts of behaviors seen in the data. The setting that we consider in this set of experiments is one where the agent utilizes diverse prior data to solve a task, even when the prior data does not actually succeed at solving it. Offline RL has shown success at generalizing behaviors to new tasks in robotic manipulation [28], but not yet to improve human behavior.

We have a very simple goal: use offline RL to learn a policy that knows how to influence humans behaving suboptimally to take better actions. As described in Section 3, we use CQL [18] as our offline RL algorithm due to its effectiveness in reducing distribution shift. We defer implementation details, *i.e.*, architecture and hyperparameter choices, to Appendix A.

One of the main benefits of offline RL over imitation learning is its ability to leverage diverse prior data to generalize to solving new tasks, even when the data does not contain any complete examples for the new tasks [28, 19, 35]. In our work, we also collect and train on a multi-objective dataset, but unlike prior works, our dataset consists entirely of episodes where humans interact with other humans. Different objectives are created by changing the reward function of the environment. Such data can be used for generalization by simply relabeling the rewards to match that of the new task.

### 5.1 Experiment

**Task description.** We empirically evaluate our approach in the *Overcooked* domain. We consider two different layouts, which we call *Asymmetric Advantages* and *Forced Coordination* (Figure 3), that often require coordination in the form of passing ingredients to improve efficiency. The RL-controlled agent is the one with the blue hat.

Most prior works on *Overcooked* both train and evaluate on the standard task [6, 20]. In this work, we want to demonstrate that offline RL can accomplish something more complex – we can train an agent from multi-task data to solve new tasks by influencing its human partner in clever ways, even when *the prior data contains no evidence of intentional and successful*

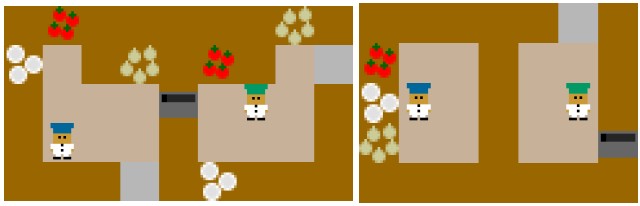

Figure 3: Layouts *Asymmetric Advantages* (left) and *Forced Coordination* (right).

*influence for the new task*. Our new tasks use a modified reward function, *e.g.*, soups with tomatoes are more valuable to the customers than with onions, such that humans who are optimizing the standard reward (which rewards any type of soup equally) behave very suboptimally. Due to this suboptimality, it is important for the learned agent to influence its human partner, *e.g.*, to pick up tomatoes instead of onions. We evaluate on two new tasks using the aforementioned layouts: *Asymmetric Advantages (Human Deliver)* offers doubled reward if the soup is delivered specifically by the human partner of the ego agent, and *Forced Coordination (Tomato Bonus)* offers doubled reward if the soup contains only tomatoes.

**Data collection.** We collected a dataset of human-human play where the human players were provided with one of several different instructions, in order to gather a diverse dataset that illustrates a variety of behaviors and human-human interactions. In the first set of instructions, humans are playing the game under the standard objective. We collected 20 human-human trajectories of length $H = 1,200$. In the second, humans are playing the game but one of the humans is given the additional task of moving ingredients to counters for a small reward; interestingly, the other human is not aware of this objective, which induces various suboptimalities because they cannot predict their partner's actions. Under this modified objective, we collect 5 human-human trajectories of length $H = 1,200$.

**Baselines.** We compare offline RL as described in Section 5 to two baselines that also do not assume access to online interactions, nor simulators of the environment: naive behavior cloning (BC) on the dataset, and filtered BC that takes the subset of 10 trajectories that result in the highest reward. We also include a comparison to Fictitious Co-Play (FCP) that assumes an environment simulator and is a common baseline in collaborative tasks [29]. FCP trains a population of randomly-initialized agents via self-play.

**Evaluation.** We deployed each policy alongside a human player. We recruited 10 different human users for this evaluation. For each layout and method, we collected 30 trajectories of length $H = 400$.

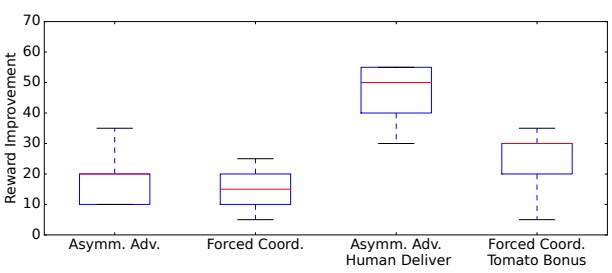

Figure 4: Improvement in reward (taken as difference in cumulative reward between last 100 and first 100 timesteps) when a human is paired with our learned agent (via offline RL) across 30 trajectories of $H = 400$ timesteps.

In Figure 5, we reported the mean and standard error of reward across all trajectories when solving both the standard task, as well as generalization to the new ones. We see that across all domains, offline RL outperformed both BC and filtered BC, as well as FCP. This is because both BC and filtered BC cannot exceed human-human performance, which is often suboptimal due to imperfect allocation of tasks for coordination. The difference is exaggerated when evaluating generalization, where human-human performance in the dataset solves different tasks. In addition, self-play approaches fail when the human partner is strategically suboptimal, whereas our learned agent via offline RL can account for that by influencing the partner to change their strategy.

Next, we ask whether this improvement in reward is solely due to the agent performing well on their part of the task, or due to an actual impact on human behavior. Qualitatively, in Figure 6, we show that offline RL is able to learn a sensible influencing strategy to influence the human partner to plate

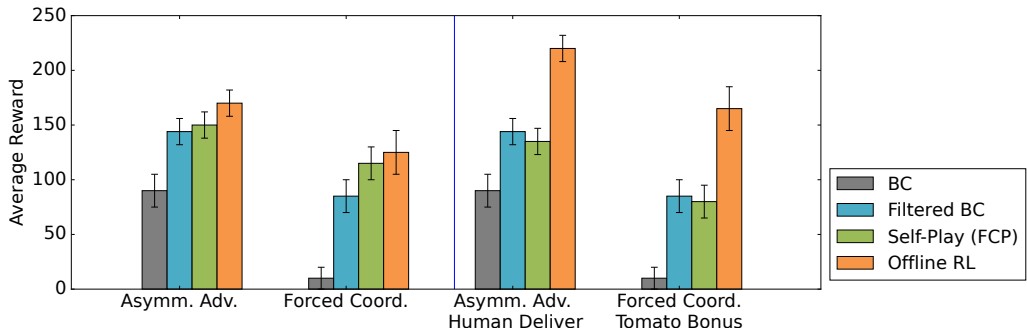

Figure 5: Average rewards over 30 trajectories of $H = 400$ timesteps of learned agents paired with real humans, with standard error across humans.

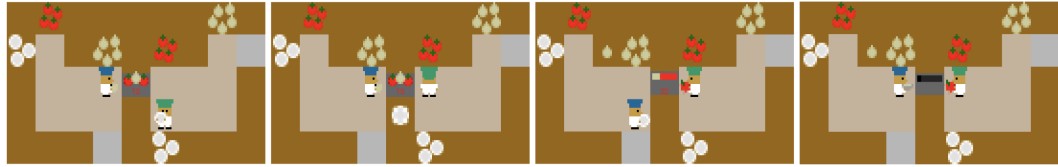

Figure 6: Trajectory where the ego agent influences their partner to plate the soup by moving a plate on the counter nearby. The partner ends up dropping their onion in order to plate the soup. This leads to more optimal coordination as the partner is in a more advantageous position to deliver soups.

the dish, which is crucial for success in *Asymmetric Advantages (Human Deliver)*. In the visualized trajectory, the learned agent moves a plate near the human and leaves to pick up ingredients for the next soup; the human realizes this and plates the soup on the stove. We verify that this behavior does not appear at all in the prior data. However, components of this behavior (such as mistakenly picking up a plate and subsequently placing it on a counter) appear in the prior data. In addition, in Figure 4, we show that the mean reward achieved in the last 100 timesteps greatly exceeds that of the first 100 timesteps, suggesting that coordination is improving over time between our agent and a human.

## 6   Achieving Long-term Influence of Latent Strategies

In this section, we consider how offline RL can learn to effectively achieve long-term influence on the human's policy itself. Accomplishing this type of adaptive influence requires knowledge of what the other human intends to do, or their *latent strategy*. In contrast to Section 5, we are not trying to show that offline RL can learn influence strategies to affect a human's actions that do not appear in the data, but rather that offline RL can influence a human's policy, while adapting to changes in the policy.

We propose an offline RL method that learns a low-dimensional representation of the human's latent strategy, and condition the learned policy's behavior on that strategy. The method we propose involves separately learning such representations, then a policy that conditions on these representations as part of its state. We describe the high-level approach but defer implementation details to Appendix A.

### 6.1   Learning Representations of Latent Strategy

First, we describe how we learn representations of latent strategy. Recall from the definition of Hi-MDPs in Section 3 that the other human's behavior follows latent dynamics $P^z(\mathbf{z}' \mid \mathbf{s}, \mathbf{a}, \mathbf{z})$, where $\mathbf{z}, \mathbf{z}' \in \mathcal{Z}$ are representations of the human's latent strategy; we use $d$-dimensional embeddings as the representations, with $d$ as a hyper-parameter.

Additionally, we introduce $h$ to be the agent's *history* of previous $c$ steps of interactions, where $c \in \mathbb{N}$ is a tunable parameter. Namely, let $\tau = \{(\mathbf{s}_1, \mathbf{a}_1), \ldots, (\mathbf{s}_H, \mathbf{a}_H)\}$ denote a trajectory taken by the ego agent, then we denote the history at step $i$ corresponding to state $\mathbf{s}_i$ and action $\mathbf{a}_i$ as

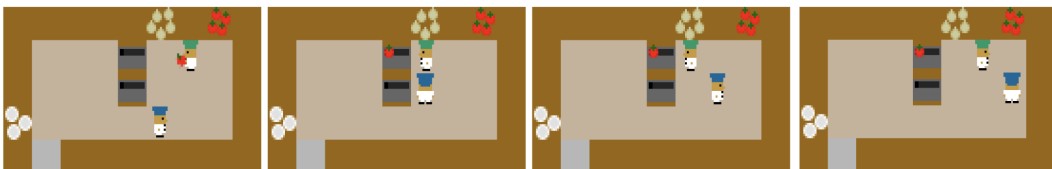

Figure 7: Example trajectory where the ego agent influences their partner to pick up tomatoes rather than onions by blocking off access to the onions. Note that the agent should only block their partner if they believe their partner is going towards the onions. Once their partner appears to be going to the tomatoes instead, the agent returns to their own task.

$h_i = \{(\mathbf{s}_{i-c}, \mathbf{a}_{i-c}), \ldots, (\mathbf{s}_{i-1}, \mathbf{a}_{i-1})\}$. Note that it is easy to augment our offline dataset such to now include such histories $\mathcal{D} = \{\mathbf{s}, \mathbf{a}, r, \mathbf{s}', h\}$.

We approximate the latent dynamics by learning an encoder $f(\mathbf{z}|h)$ that uses the history of interactions to predict a distribution over representations of the human's latent strategy. Note that we use history instead of the human's previous strategy because we anticipate that history has enough information to reconstruct the latter. This greatly simplifies training the encoder. In practice, the encoder can be of the form $f(\mathbf{z}|h) = \mathcal{N}(\mathbf{z}; f_\mu(h), f_\Sigma(h))$, where $f$ is a neural network that outputs both the $d$-dimensional mean of $\mathbf{z}$ as well as the $d \times d$ covariance matrix.

Although we do not observe the ground truth representations of latent strategy, we can still learn these representations in an unsupervised way, since we know that they should be predictive of future states. Recall that the dynamics of Hi-MDPs are given by $P(\mathbf{s}'|\mathbf{s}, \mathbf{a}, \mathbf{z})$. We propose jointly training a decoder $d(\mathbf{s}'|\mathbf{s}, \mathbf{a}, \mathbf{z})$ that estimates the transition dynamics and reconstructs the next state. We ultimately propose the following joint training objective:

$$\arg\max_{f,d} \mathbb{E}_{\substack{\mathbf{s},\mathbf{a},\mathbf{s}',h\sim\mathcal{D}, \\ \mathbf{z}\sim f(\mathbf{z}|h)}} [\log d(\mathbf{s}' \mid \mathbf{s}, \mathbf{a}, \mathbf{z})] - \mathbb{E}_{h\sim\mathcal{D}} [D_{\mathrm{KL}}(f(\mathbf{z} \mid h) \parallel p_0(\mathbf{z}))] . \tag{1}$$

The first term in (1) is the reconstruction loss of the next state, and the second term is an information bottleneck [2] that regularizes the encoder towards prior $p_0(\mathbf{z})$. In our work, we use predicted distribution for the previous timestep as the prior, starting with $p_0(\mathbf{z}) = \mathcal{N}(0, I)$. By doing so, the inferred representations from the encoder are encouraged to be sequentially consistent across time, discouraging abrupt changes in behavior. Note that Xie et al. [34] propose a similar objective to model other agents in the environment, but only consider the reconstruction loss because each agent's latent strategy is fixed within an episode.

## 6.2 Offline RL Conditioned on Latent Strategy

Recall the example of directing the human towards picking up tomatoes, where the agent's strategy to influence the human depends on how the human behaves. This naturally means that we should condition the agent's policy on the human's latent strategy. We therefore train a policy parameterized as $\pi(\mathbf{a}|\mathbf{s}, \mathbf{z})$, where $\mathbf{z} \sim f(\mathbf{z}|h)$ is the representation of latent strategy sampled by the encoder $f$ from Section 6.1. By doing so, the agent utilizes the learned dynamics of human strategy, and reasons how to influence the human's strategy towards behaviors that better maximize reward.

Our algorithm is a modification of CQL. Namely, because the learned policy $\pi$ additionally conditions on human behavior, so does the Q-function $Q(\mathbf{s}, \mathbf{a}, \mathbf{z})$. Our modified actor-critic algorithm learns Q-functions via an objective similar to CQL, but where $\mathbf{z}$ is sampled from the encoder, and serves as an additional input to the Q-function and policy.

## 6.3 Experiment

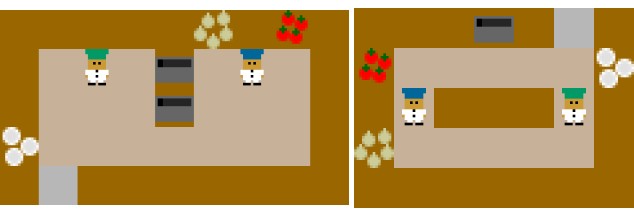

Figure 8: Layouts *Open Asymmetric Advantages* (left) and *Counter Circuit* (right).

**Task description.** We again evaluate our described approach in the *Overcooked* domain. We consider a different set of layouts, which we call *Open Asymmetric Advantages* and *Counter Circuit* (Figure 8). In contrast to the layouts in Figure 3, these layouts allow for players to block each other,

which we view as an influence strategy that requires knowing the partner's behavior. We consider the modified task where a soup that is delivered containing only tomatoes yields double the reward of other soups. We do so because this encourages influence by the ego agent on their human partner. If the partner appears to be going to pick up onions, the agent should try to redirect their focus (*i.e.*, by blocking the path towards the onions).

**Data Collection.** For each layout, we collect a dataset of 20 human-human trajectories of length $H = 1,200$. We inform the humans that is the ego agent about the modified objective, and let the other human know that they are missing information. This biases the ego agent to try various influence strategies, and for their partner to respond accordingly.

**Baselines.** We compare our proposed behavior-conditioned offline RL approach, which we call *latent offline RL*, to several baselines. The first two are standard BC and offline RL, which simply condition on the current state and do not try to infer the human's behavior. The remaining baseline, which we call *memory offline RL*, implicitly conditions on the human's behavior by conditioning on a history of states. The difference between this baseline and what we propose is that *memory offline RL* does not have an additional objective to learn representations of latent strategy as we propose in Section 6.1. Our final baseline is an offline variant of the LILI algorithm [34], which we dub *offline LILI*. We choose LILI as a baseline because they similarly model human behavior by learning latent representations. However, LILI assumes (1) access to online data and an environment simulator and (2) that changes in behavior solely happen between episode. Thereofre, we modify LILI to match our problem statement as follows: (1) we discard the recollection of data steps in the LILI algorithm, and (2) we re-estimate the human strategy every $c$ timesteps within an episode.

**Evaluation.** We deployed each policy alongside a human player, using the same 10 human users for the evaluation in Section 5.1. We collected on 30 trajectories of length $H = 400$.

In Figure 10 (left), we report the mean and standard error of reward across all trajectories in both layouts when the learned agents are paired with human partners. We notice that taking into account human behavior in some way is beneficial, as both *memory offline RL* and *latent offline RL* greatly outperformed naive BC and offline RL. Surprisingly, our approach (*latent offline RL*) noticeably improved upon *memory offline RL*, even though both were given the same amount of information. We hypothesize that disentangling the learning of human behavior and optimal influence leads to more effective training when the policies are neural networks. Our method also improves greatly over *offline LILI* due to modeling more nuanced transitions in behavior.

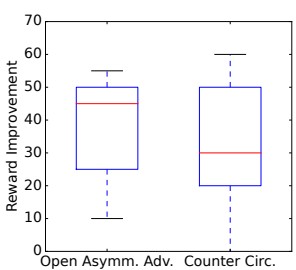

Figure 9: Improvement in reward (taken as difference in cumulative reward between last 100 and first 100 timesteps) when a human is paired with our agent (latent offline RL).

In addition, we provide evidence that when agents are successful, it is due to the human partner changing their behavior. In Figure 10 (right), we evaluate all the considered agents when paired with a scripted policy that unconditionally chooses the suboptimal action, *e.g.*, picking up onions when tomatoes are superior, and cannot be influenced to do otherwise. We see that since all evaluated approaches roughly perform the same, we rule out the possibility that agents learned via our proposed approach are solving tasks more successfully in isolation. Then, in Figure 9, we show that the mean reward our learned agent achieves with a human partner in the last 100 timesteps greatly exceeds that of the first 100 timesteps; since our learned agent does not change behavior unless their partner does, this shows that the human partner was successfully influenced towards more optimal coordination. Qualitatively, in Figure 7, we show that our proposed *latent offline RL* learns an adaptive influencing strategy to influence the human partner to pick up tomatoes in the *Open Asymmetric Advantages* layout. In the trajectory, the learned agent blocks access to the onions, only until it infers that the human is not going for them.

## 7 Discussion

Humans often behave suboptimally, either due to systematic biases or incomplete knowledge of the environment. In this paper, we investigate whether offline RL on human-human interactions is capable of learning policies that can influence humans toward more desirable behavior. First, we showed that given a dataset with enough variability in human behaviors, existing model-free offline

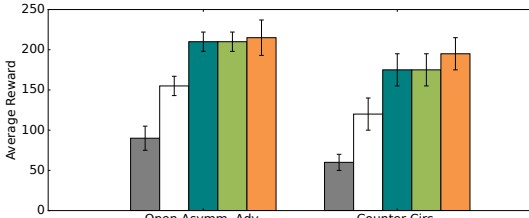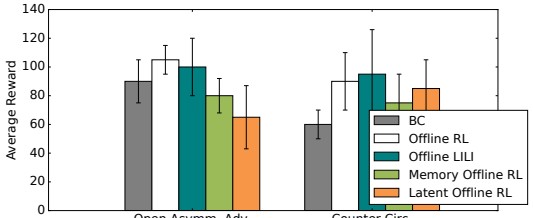

Figure 10: Average rewards over 30 trajectories of $H = 400$ timesteps of learned agents paired with real humans (left) and with a scripted policy that will always pick up onions (right). Note that the scripted policy cannot be influenced.

RL algorithms can learn to intentionally perform new influence strategies that are not explicitly performed by the humans in the dataset. However, humans will often adapt their policy based on past experiences, meaning agents need to recognize how the human is currently behaving, or their latent strategy, and then adapt influence strategies to account for changes in behavior. Therefore, we also proposed an offline RL algorithm that estimates and conditions on latent strategy, and show that it can successfully influence humans even when their latent strategy changes.

**Limitations and future work.** We focus our evaluation on the *Overcooked* game, because it has interpretable rules and dynamics, while capturing the fundamental challenges to learning human influence. However, it is still substantially simpler than real-world applications that require collaboration with humans. An important direction of further investigation is to investigate whether our findings hold in more realistic domains (*e.g.*, dialogue systems).

In this work, we separately consider two challenges of human influence: learning unseen influence strategies, and how to achieve long-term influence that co-adapts with changes in human policies. It remains to be seen if offline RL can solve both challenges simulataneously, in one end-to-end approach – from diverse human-human interactions, learn new strategies to influence human actions, then extend these strategies (potentially in simulation or online) to achieve long-term influence.

**Ethical implications.** Perhaps most importantly, the notion of influence and "improvement" of human behavior is a double-edged sword. On the one hand, humans behavior is suboptimal and people might lack knowledge that agents have, and so agents can serve implicitly as coaches that help people arrive at better strategies. On the other hand, when agents are themselves lacking knowledge (especially of human values and preferences), influencing people to perform "better" under the wrong objective or wrong world state can very much hinder their performance. This is painfully obvious in recommender systems, where influence with the wrong objective leads to much wasted time on social medial platforms and, worse, polarization and other negative societal effects. Endowing the agent with proper uncertainty is a crucial next step for this line of work.

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
