# OpenReview forum: "Learning to Influence Human Behavior with Offline Reinforcement Learning"
_NeurIPS.cc/2023/Conference — NeurIPS 2023 poster_

### Official Review · Reviewer_hp68 · 2023-06-27

**Soundness:** 1 poor
**Presentation:** 3 good
**Contribution:** 2 fair
**Rating:** 6
**Confidence:** 3

**Summary:**

This paper presents an investigation for learning to influence suboptimal human opponents in agent-human interactions. It claims that by performing offline RL on human-human interaction data, an agent can learn to influence its opponent's actions and latent strategies, even if there is no explicit information of influence in the data. In the first set of experiments, the authors showed that CQL achieved higher test returns when paring with human opponents, and in the second set of experiments, they showed that decoding latent representation for states performed better than two other opponent modeling technique.

**Strengths:**

1. The paper is well-written. Materials are organized in a way that facilitates reading, and the connection to existing literature is clearly stated.
2. Learning to influence human opponents, especially those suboptimal ones, is an important problem for human-agent interactions.
3. The idea of learning influence strategies from human-human data is interesting and intuitive.

**Weaknesses:**

1. There lacks explicit empirical evidence to support the claims made in this paper.

The authors claimed that: (a) agents can learn to influence their opponents' actions using offline RL (the CQL algorithm), and (b) agents can learn to influence their opponents' latent strategies. However, there is no direct quantitative evidence of the increase in such influence. The reported improvement in test return can be explained by many factors. For example, rather than influencing its opponent, an agent may proactively solve the task itself. The only evidence is four states of a trajectory, yet it is not sufficiently annotated for readers.

2. The presentation for modeling latent strategies is confusing.

This paper does not offer a tangible description of the "latent strategies". From eq (1), the "latent strategies" seem to be vectors that improve the prediction of the next states. But eq (1) seems not characterize the relationship between these vectors and the actions of the opponent. So I agree that they are latent vectors that improve policy learning, but I cannot see why they are strategies of human opponents.

**Questions:**

1. Since this paper considers a multi-agent setting, I wonder how the MDP formulation in sec 3 covers the information of the opponents. In particular, do you consider the opponent's actions as part of the states or part of the actions?

2. Improvements in test return can be explained by multiple factors. To explicitly show that offline RL shapes human opponents' actions or strategies, more explicit evidence is necessary. For example, what about reporting the action frequencies of humans?

**Limitations:**

Yes, they have.

---

> ### Author Rebuttal · Authors · 2023-08-10
>
> Thank you for your review. The main concerns in your review seem to center around the evidence to support our conclusions. We hope to remedy this by providing additional evaluations that further support our hypothesis that the human does improve their behavior during evaluation (implying successful influence) and that the learned agent is not simply more adept at solving the task independently. We go into these additions in more detail below, and are happy to compute any others that you may suggest. Please let us know if these additions fully address the issues in your review.
>
> **No direct quantitative evidence of influence**
>
> You raised a good point that the improved performance may not be due to influence. To remedy this, we report the following additional metrics on the evaluation runs:
>
> 1. In addition to showing the cumulative reward obtained, we will show the difference in cumulative reward obtained between the first and last 100 timesteps of each evaluation. As shown in Figure 1 (of the attached file), the reward obtained towards the end of evaluation is much higher than in the beginning. Since the learned agent only changes behavior when the human partner does, this means that the human must have adapted their behavior over the course of evaluation.
>
> 2. To rule out that the learned agent is not simply “solv[ing] the task itself”, we evaluate all the considered approaches against a scripted policy that always performs the suboptimal action we would expect from a naive human (such as grabbing onions when tomatoes yield higher reward). In Figure 2, we see that the learned agent using our approach actually performs worse than naive offline RL. This means that our learned agent cannot be solving the task more successfully independently, but rather is trying to influence its human partner.
>
> If you have ideas for other metrics that we can compute to support our conclusion further, we are happy to include them in the updated paper.
>
> **Characterize the relationship between latent strategies and the actions of the opponent**
>
> We agree that this would be informative. We attempt to do so in Figure 3, where we plot several latent strategies inferred from trajectories during evaluation (using PCA to reduce to 2 dimensions). We then colored the points that correspond to trajectories where the human partner goes to get tomatoes as red, and ones where the human goes to get onions as green. The plot shows that the red and green points are prominently in different clusters, showing that the latent strategies correspond to different high-level actions taken by the human.
>
> **Answers to questions**
>
> You also raised questions that we aim to answer below:
>
> 1. In this work, we consider the partner’s actions as part of the state, since they are observed by the learning agent. The partner’s intentions/strategy, however, are unobserved, making our MDP a special case of a POMDP. We will clarify this in the updated paper.
>
> 2. We agree that more evidence is necessary. We propose adding the following metrics described above, and shown in the attached file. We are happy to include any other metrics you might be helpful to support our conclusion.

---

> > ### Comment · Reviewer_hp68 · 2023-08-15
> >
> > Thanks for your responses. I have adjusted my evaluations accordingly.

---

### Official Review · Reviewer_q3SJ · 2023-07-05

**Soundness:** 2 fair
**Presentation:** 3 good
**Contribution:** 3 good
**Rating:** 5
**Confidence:** 4

**Summary:**

This paper presents an investigation into the application of offline RL on a dataset of human-human interactions to develop a policy capable of influencing human behaviors towards desired outcomes. The authors demonstrate that the learned policy not only affects immediate behavior but also influences the long-term strategies and preferences of the human partner. Experimental results provide strong evidence for the effectiveness of offline RL in learning to influence humans and achieving desirable rewards.

**Strengths:**

1. This paper is well-written and easy to follow.

2. The utilization of offline RL on an existing dataset of human interactions is a straightforward yet effective approach.

3. The problem of learning to influence human behaviors from existing datasets is both interesting and important for the field

**Weaknesses:**

1. The experiment solely employs offline RL methods as baselines. It would be beneficial to compare offline RL methods with approaches that leverage both offline data and online interactions. For example, one potential baseline could involve first acquiring a learned human model via behavior cloning and subsequently learning an adaptive policy to cooperate with the learned human model in the simulator, as demonstrated in [1].

2. The evaluation appears to be limited to assessments involving human participants. It would be valuable to evaluate the learned agent against a learned human model or scripted policies, as demonstrated in [1] [2].

3. The claim that there is no evidence of influence between humans, coupled with the assertion that the learned policy can induce long-term changes in human strategy, may be overly conclusive and requires further substantiation.


Reference:
1. Strouse, D. J., et al. "Collaborating with humans without human data." Advances in Neural Information Processing Systems 34 (2021): 14502-14515.
2. Yu, Chao, et al. "Learning Zero-Shot Cooperation with Humans, Assuming Humans Are Biased. ICLR 2023.

**Questions:**

1. The authors state that the dataset contains no evidence of influence. However, how can one differentiate between behaviors resulting from mistakes and those intentionally performed by the human player to influence their partner?

2. The data collection process outlined in Section 5 involves various approaches. What impact does the dataset have on the learned policy? Would the learned policy differ significantly if only the first or second set of instructions were used for training?

3. Section 6 claims that the learned policy can induce long-term influence on the latent strategies of the human partner. Besides the average episode reward, is there any additional evidence supporting the assertion that the learned policy genuinely influences humans in the long term? The case study presented in Figure 6 and Lines 390-393 demonstrates short-term behavior influence, where the agent blocks humans from picking up onions to encourage tomato selection. My question is whether this influence is truly long-term, meaning, would the human player exhibit a stronger preference for picking tomatoes after repeated instances of being blocked from picking onions by the learned agent?

Clarity issues:
1. In Line 208, the term "naive offline RL" is not immediately clear. Please provide a more precise explanation.

2. Regarding Figure 5 and Figure 6, it would be better to mark which chef is controlled by the human player and which chef is controlled by the learned policy.



**Limitations:**

This work currently focuses on relatively simple environments. It would be promising to scale the method to more complicated systems.

---

> ### Author Rebuttal · Authors · 2023-08-10
>
> Thank you for your review. You raised some interesting additional evaluations to strengthen our paper that we have added and discussed below. Please let us know if our response fully addresses the issues in your review.
>
>
> **Compare with approaches that leverage both offline data and online interactions**
>
> In this work, we considered methods that were purely offline (can only use human-human data) and did not have access to an environment or human simulator. We believe that this limiting setting is important to study to find approaches that work in complex, real-world settings.
>
> However, we agree that many existing approaches in the broader domain of human-AI collaboration use self-play, thus circumventing requiring a human simulator (though they still require a simulator of the environment/game, which is often easier to obtain).  We believe that self-play approaches fail when the human partner is strategically suboptimal, and the learned agent can account for that by influencing the partner to change their strategy.  Though Yu et al. consider biased humans, they do not consider that human behavior can change within a trajectory, thus not allowing for influence like we do. In Figure 4 (of the attached file), we reported results for an agent learned via self-play (FCP), and see that it does not perform as well as ours when the human has a misaligned objective.
>
>
> **Evaluate the learned agent against a learned human model or scripted policies**
>
> We believe that it is important to evaluate our work against real humans because they exhibit nuances that traditional simulators typically do not emulate. Namely, real humans are able to adapt their behavior, which allows for such behavior to be influenced by changes in state or the actions of other agents. However, we see that evaluating against simulated, stationary policies can still be useful to show that our approach actually works due to influence. To rule out that our improved performance is simply due to solving the task better independently, we evaluate all the considered approaches against a scripted policy that always performs the suboptimal action (such as grabbing onions when tomatoes yield higher reward). In Figure 2, we see that the learned agent using our approach actually performs worse than naive offline RL. This means that our learned agent cannot be solving the task more successfully independently, but rather is trying to influence its human partner.
>
>
> **Claims require further substantiation**
>
> For the claim that there is no evidence of influence between humans, we are not referring to influence of any kind, but specifically influence that assists in solving the new generalization task. Specifically, in the new task where one of the humans receives higher reward for plating, one form of influence (that offline RL learns) is to pass the plate to the human to encourage them to plate. We can confirm that this influence strategy does not appear in the offline data at all by simply inspecting all the trajectories (as there are only 25).
>
> For the claim that there is long term influence, we provide the following quantitative evidence. We show the difference in cumulative reward obtained between the first and last 100 timesteps of each evaluation. As shown in Figure 1, the reward obtained towards the end of evaluation is much higher than in the beginning. Since the learned agent only changes behavior when the human partner does, this means that the human must have adapted their behavior over the course of evaluation i.e. the human knows to pick tomatoes over onions.
>
>
> **Answers to questions**
>
> The reviewer also raised questions that we aim to answer below:
>
> 1. We hope to have clarified what we meant by no evidence of influence above. Regarding accidental vs intentional influence, we believe that both would result in the same actions being observed in the data i.e. passing a plate looks the same even if done accidentally. Because we did not observe such actions in the dataset, we can confidently say that no accidental or intentional influence appeared.
>
> 2. This is an important point, as offline RL cannot come up with entirely new strategies from scratch. Our hypothesis is that the dataset must demonstrate a reasonable range of behaviors such that such behaviors can be stitched to produce a new strategy. In the example of passing a plate, offline RL cannot learn to do this if there is no movement of objects at all in the dataset, but can if there is evidence of humans moving to and picking up objects from shared tables. We ensured that our dataset had these components.
>
> 3. We hope that our additional evaluation shows evidence of long-term influence. If you have others in mind, we would be happy to compute and include them in the updated paper.

---

> > ### Comment · Reviewer_q3SJ · 2023-08-18
> >
> > Thanks for your responses. I would maintain my original scores.

---

### Official Review · Reviewer_rRJq · 2023-07-10

**Soundness:** 2 fair
**Presentation:** 3 good
**Contribution:** 2 fair
**Rating:** 4
**Confidence:** 3

**Summary:**

This work aims to provide some evidence that offline reinforcement learning techniques can be used in the field of human-agent collaboration to influence or improve the behavior and underlying strategies of humans. The authors first verified that agents trained by CQL can influence the behavior of human players in some scenarios. Then, by simply modifying the CQL by conditioning on human latent strategy representations, the agents can learn to adapt to changes in human behaviors.

**Strengths:**

The research purpose of this work is an important topic in the field of AI, how to improve human performance in human-agent collaboration. The author's attempt to verify the effectiveness of offline reinforcement learning techniques in human-agent collaboration scenarios is commendable.

**Weaknesses:**

1. After carefully reviewing the manuscript, I personally think the current content (methods and experimental results) does not effectively demonstrate the thesis that offline RL can learn to guide humans toward better performance by combining human latent policy representations. I agree that the agent may learn to adapt to changes in human behavior through offline RL, but it remains unclear how offline RL can tangibly influence and improve human performance. Furthermore, there is a wealth of research in the field of human-agent collaboration, such as Du, et al.[1] and Alamdari, et al.[2], focus on developing assistive agents to improve human performance. It would be advantageous to discuss this research in the paper to provide a more comprehensive understanding of the field. In the experimental part, the existing experimental results (improved team rewards & few examples) cannot support this assertion either. I suggest that the authors provide some objective metrics of human participant performance to make the conclusion more solid.

2. User Study is crucial in human-agent collaboration research, and I’d suggest the authors provide additional information, including:
     - Ethical review: Did participants provide informed consent for their involvement in this research and for the use of their data toward this project?  Were they fully informed about the purpose of the research, did they confirm their approval for involvement, were they told about how to withdraw their data?
     - Test settings: What is the proficiency of the participants in the game? Were they all novices or professionals? Whether to provide a standard test specification or guide before the test to ensure the consistency of the test purpose?

3. Recent research in the field of human-agent interaction has highlighted the significance of evaluating agents using both objective and subjective metrics. See Strouse, et al.[3], and McKee, et al.[4]. Incorporating subjective metrics into the evaluation of agents can provide a more holistic understanding of their impact on human performance and well-being. For example, did participants prefer playing with the trained agent over other baseline agents? During the collaboration, did the participants perceive their actions to be influenced by the agent's behaviors? If yes, to what extent? Did all participants perform better as a result?

4. This work attempts to verify the effectiveness of existing offline RL techniques in some human-agent collaboration scenarios. Due to the lack of comparisons with state-of-the-art methods in the field of human-agent collaboration, such as Strouse, et al.[3], Yu, et al.[5], etc., it remains unclear where the boundary of offline RL in human-agent collaboration is.

5. There are some unevidenced and unsubstantiated claims in the manuscript. For example,
	- In line 208, "It is evident that naive offline RL cannot learn adaptive policies."
	- In line 179, "humans are passive and will often only respond to what their partner is doing"

6. Minor Concerns:
	- Many references are not uniformly formatted, such as [27] "nature" and "Science" ,  whether conference abbreviations, e.g.: "(ICLR)", are reserved.
	- In line 369,  "Thereofre" should be "Therefore"
	- In line 412, "simulataneously" should be "simultaneously"

[1] Du, et al. Ave: Assistance via empowerment. 2020.

[2] Alamdari, et al. Be considerate: Avoiding negative side effects in reinforcement learning. 2022.

[3] Strouse, et al. Collaborating with Humans without Human Data. 2021.

[4] McKee, et al. Warmth and competence in human-agent cooperation. 2022.

[5] Yu, et al. Learning zero-shot cooperation with humans, assuming humans are biased. 2023.

**Questions:**

See the revision recommendations in the "Weaknesses" section.

**Limitations:**

As discussed above, this work lacks the comparison of SOTA methods, and the user studies are not sufficient, which would limit its reliability.

---

> ### Author Rebuttal · Authors · 2023-08-10
>
> Thank you for your review. The main concern raised in the review seems to center around the evidence that supports our conclusion. We hope to remedy this by providing additional evaluations that further reinforce our hypothesis that the human does improve their behavior during evaluation trials (implying successful influence), and offer some interpretability regarding the latent representations. We go into these additions in more detail below, and are happy to compute any others that you may suggest. You also asked for more details regarding the evaluation and discussion with related works that we offer below and will include in our updated paper. Please let us know if our response fully addresses the issues in your review.
>
> **Current content does not effectively demonstrate influence**
>
> We understand your viewpoint that the improved performance may not be due to successful influence. Therefore, we provide the following metrics to make our conclusion more solid. In our opinion, these are the most reasonable metrics we could compute using our current evaluation data:
>
> 1. In addition to showing the cumulative reward obtained, we show the difference in cumulative reward obtained between the first and last 100 timesteps of each evaluation. As shown in Figure 1 (of the attached file), the reward obtained towards the end of evaluation is much higher than in the beginning. Since the learned agent only changes behavior when the human partner does, this means that the human must have adapted their behavior over the course of evaluation.
>
> 2. To confirm that the learned agent is actually trying to influence its partner, rather than solve the task itself, we evaluate all the considered approaches against a scripted policy that always performs the suboptimal action we would expect from a naive human (such as grabbing onions when tomatoes yield higher reward). In Figure 2, we see that the learned agent using our approach actually performs worse than naive offline RL. This means that our learned agent cannot be solving the task more successfully independently, but rather is trying to influence its human partner.
>
> 3. To further support our hypothesis that using latent representations of strategy helps performance, we additionally show that the learned latent strategies map to different high-level actions. In Figure 3, we plot several latent strategies inferred from trajectories during evaluation (using PCA to reduce to 2 dimensions). We then colored the points that correspond to trajectories where the human partner goes to get tomatoes as red, and ones where the human goes to get onions as green. The plot shows that the red and green points are prominently in different clusters.
>
> If you have ideas for other metrics that we can compute to support our conclusion further, we are happy to include them in the updated paper. Your suggestion to include user feedback regarding which learning agent they preferred is very useful. However, this requires redoing our evaluation, which we could not do in the rebuttal but would be happy to do in the future if you feel it would strengthen our evidence.
>
> **Discussion with other related works**
>
> Thank you for bringing up these additional papers. We will include a discussion of them in our updated paper. Though the papers also look at coordination with humans, we view our paper as tackling a different problem, in that we consider influence rather than assistance. In the latter, the learned agent will always try to support the human’s intentions or strategy, whereas in our work, the learned agent tries to change the human partner’s strategy if it is suboptimal. Therefore, the methods proposed in the works you referenced would not be applicable in our setting, where the humans are not only suboptimal due to unfamiliarity with controls, but also due to having the wrong idea of what high-level action to take.
>
> **Additional details on user study**
>
> Thank you for raising this important concern. We provide the additional details below and will include them in the updated paper.
>
> Ethical review: Our user study is IRB-approved. The participants did provide consent and were informed about the high-level goal (of influence during coordination tasks) but not of the methodology that we used or our specific hypothesis (that pure offline RL can achieve successful influence). Per our IRB protocol, there is no option to withdraw the unidentified data.
>
> Test settings: The participants are not familiar with our specific game setting, though our controls are fairly intuitive and we provide detailed instructions on how to play. We found that the participants quickly adapted to the unfamiliar controls and learned to pilot their characters effectively. Therefore, much of the suboptimality came from not knowing the optimal high-level action to take.
>
> **Lack of comparisons with state-of-the-art methods**
>
> In this work, we considered methods that were purely offline (can only use human-human data) and did not have access to an environment or human simulator. We also specifically study the setting of influence in human-AI collaboration. In this setting, we believe that the Offline LILI baseline is the current state-of-the-art approach.
>
> However, we do think that the self-play approaches you referenced, though they assume an environment simulator, are interesting to study as they are commonly used to solve general collaboration tasks. We believe that self-play approaches fail when the human partner is strategically suboptimal, and the learned agent can account for that by influencing the partner to change their strategy. Though Yu et al. consider biased humans, they do not consider that human behavior can change within a trajectory, thus not allowing for influence like we do. In Figure 4, we reported results for an agent learned via self-play (FCP), and see that it does not perform as well as ours when the human has a misaligned objective.

---

> > ### Author Response · Authors · 2023-08-17
> > **Let us know if you have any additional concerns**
> >
> > Thank you for your review and comments. We hope that our additional evaluations and rebuttal have addressed your primary concerns with our paper. We would really appreciate feedback as to whether there are any (existing or new) points we have not covered, and we would be happy to address/discuss them!

---

> > ### Comment · Reviewer_rRJq · 2023-08-17
> >
> > I believe that detailed and comprehensive user studies are crucial to support the authors' claim that "agent can influence and improve human performance". However, the current user study results are not sufficient to support this conclusion, so I will maintain my recommendation.

---

### Official Review · Reviewer_RC1m · 2023-07-28

**Soundness:** 4 excellent
**Presentation:** 4 excellent
**Contribution:** 3 good
**Rating:** 6
**Confidence:** 4

**Summary:**

This paper proposes a novel framework to empower an intelligent agent with the capability to effectively influence suboptimal human behavior during interactions. The primary objectives revolve around tackling two key challenges: (1) deducing a new strategy to influence human action and (2) learning to influence the human's long-term latent strategy. To evaluate the proposed framework's efficacy, a set of collaborative tasks (i.e., Overcooked environment) are employed. The experimental results demonstrate that the agent can successfully learn to influence human behavior based on a human-human interaction dataset, even though no instances of successful influence were present.

**Strengths:**

Generally, I find this paper to be well-motivated, accompanied by clear and easily understandable examples. The organization and presentation of the content are well-structured, making it easy to read and comprehend. The focus of the paper lies in addressing the critical issue of enabling intelligent agent to influence suboptimal human behavior which is expected to be a topic of significant interest in the human-AI interaction research field. The proposed approach to employing offline reinforcement learning to tackle the problem is innovative such that it successfully extends the application of offline reinforcement learning in a novel context. The evaluation section is comprehensive and thorough, providing compelling evidence that supports the efficacy of the proposed method.

**Weaknesses:**

First, for the example in line 66, if we consider the real world scenarios, such behavior (i.e., repeatedly blocking the human from doing something) without providing any explanation would largely affect the human trust in the robot. This example appears to assume that the human will naturally respond by adjusting their strategy to avoid that ingredient in the future. However, this implicit assumption is quite strong and may not hold true in real-world situations.

As the authors mention in the last section, the experiments were conducted on simple simulated game-like environments rather than real world scenarios. It would be intriguing to see how the proposed approach performs when applied to more complex real-world tasks involving human-AI interactions.

Furthermore, when we talk about human-AI interaction, it includes both collaborative and competitive interactions. However, the scope of the paper is focused solely on collaborative settings, without investigating or evaluating the approach in competitive scenarios. Expanding the research to encompass competitive interactions may provide a more comprehensive understanding of the proposed method's versatility and potential applications.

Finally, in the evaluation section, the authors compare the performance between BC and offline RL methods. I feel that the paper could be further strengthened by incorporating a comparison with prior works that approximate human decision-making models and subsequently learn to influence human behavior using the learned model. This broader comparison would enhance the paper's context and shed light on the relative advantages and contributions of the proposed approach in the context of existing research.

**Questions:**

1. When the agent tries to influence human behavior, is it possible that the human gets annoyed, particularly if the influence becomes excessively insistent, dominating, or aggressive? Such a scenario could potentially lead to negative reactions from the human and undermine the effectiveness of the interaction.

2. In the provided example, the authors mention the human goes to plate when he sees the plate is right next to him. This situation seems to assume that the human has a myopic perspective. I am wondering whether the proposed approach is generalizable if the human suboptimal behavior can be attributed to other various possible reasons.

3. The agent's objective is to elicit the expected human behavior. However, the agent may generate positive action (e.g., pass the plate) or negative action (e.g., block the way). These actions may significantly impact human perception and trust. How can you ensure that the agent influences human behavior effectively without causing harm to the team?

4. Does the agent's active influence imply a leadership role, as it takes on a guiding role in shaping the collaborative decision-making process?

**Limitations:**

I appreciate the authors for the limitation and ethical implications section in the paper. Additionally, given the paper's objective of enabling the agent to influence human behavior and long-term strategies, it is essential to consider potential impacts on human trust and the possibility of unintentionally biasing the human towards a worse policy. These factors warrant serious consideration to ensure that the proposed approach fosters responsible and ethical human-AI interactions. Addressing these concerns would contribute to a more comprehensive understanding of the potential implications of the proposed method.

---

> ### Author Rebuttal · Authors · 2023-08-10
>
> Thank you for your review. We agree with you regarding expanding the scale and complexity of our evaluation. We view our work as a first-step in showing that influence can be done purely offline (without any environment or human simulators), and aim to show more compelling examples of this in the future. You also raised some interesting points that we aim to address below:
>
> **Assume that humans will naturally respond by adjusting their strategy**
>
> Note that our algorithm for influencing human strategies/policies (such as blocking an ingredient to change the person’s role in the future as well, as in the example the reviewer mentions) will only learn what appears in the data – these changes are only assumed to stick if they stick in the data, i.e. that latent transition (or similar) is observed. Of course, Offline RL can then stitch transitions in novel ways, but it will not assume a latent state / strategy transition is possible unless the data shows it. The version of our algorithm that influences human actions through state changes does not assume long-term changes in the human’s policy (but this can be unrealistic sometimes, which is why accounting for strategy change is actually needed). That said, we certainly agree that this needs to be tried out in more realistic settings. We are excited about that, and at the same time point out that this is a step far and beyond prior influence via RL work which has not studied real humans at all.
>
>
> **Expanding to competitive interactions**
>
> Though competitive settings are prevalent and interesting to study, we believe they are less relevant for influence because agents can do well by assuming the human is optimal and computing a best-response (see, for instance [cite the rohin-micah paper]). However, we agree that even in competitive settings, an agent can potentially learn to exploit a suboptimal agent. But our goal from the start has been to show how we can improve human suboptimal behavior, rather than exploit it.
>
> **Incorporating a comparison with model-based works**
>
> In this work, we considered methods that were purely offline (can only use human-human data) and did not have access to an environment or human simulator. We believe that this limiting setting is important to study to find approaches that work in complex, real-world settings.
>
> In the broader domain of human-AI collaboration, there are human-aware RL methods, which learn a human simulator from data and compute a best response via RL in an environment simulator [cite]. However, the topic of _influence_ has not been studied with these methods. Our work can be seen as paving the way towards studying influence in these settings where access to the environment model can be assumed, and our results show promise for these types of methods as well. In fact, the techniques from Offline RL that protect against distribution shift might very well apply to that setting as well, where driving the human model OOD could lead to poor real world performance.
>
> There also exist approaches that use self-play, thus circumventing requiring a human simulator and only an environment one (which is more reasonable to obtain in practice).  We believe that self-play approaches fail when the human partner is strategically suboptimal, and the learned agent can account for that by influencing the partner to change their strategy. In Figure 4 (of the attached file), we reported results for an agent learned via self-play (FCP), and see that it does not perform as well as ours when the human has a misaligned objective.
>
>
> **Answers to questions**
>
> The reviewer also raised questions that we aim to answer below:
>
> 1. Whether or not the human gets annoyed is an interesting consideration. We have not measured the users’ subjective perception in the study, but can report that whatever that perception was, it did not negatively affect the effectiveness of the interaction – we see that the collaborative task performance goes up. That said, the reviewer is 100% right that this could get annoying, and further work should examine tools to assess online whether influence is working on the individual, and perhaps adapt the type and amount of influence to the user’s response and even preferences. .
>
> 2. The myopia point is a fascinating one, thank you for bringing this up. It does seem that real people tend to be reactive in this way, and our algorithm can leverage that. But no, it does not seem that the method would be limited to myopic bias. For instance, if people have partial information about the state and that’s why they take suboptimal actions, the agent might figure out that putting certain objects in their field of view might make the user aware of them and therefore change their actions. Or more speculatively, imagine a language interaction where the agent tries to influence people to donate to a charity – there biases like a propensity to anecdotal evidence might occur, and our method would be able to pick up the fact that people behave differently once they hear such anecdotes.  Like in the first question, one might expect the effects to show up in the task performance, which the agent is optimizing. The counterargument is that not annoying users should be a first order citizen in the optimization objective – we certainly agree and think it would be useful to study reward learning applied to this type of “influence”-related preferences.
>
> 3. Yes, it is assumed that the human is suboptimal and the agent can help them overcome this, which implies the agent has better knowledge or computation than the human, putting them indeed in a leadership role. We believe when this is not true, it is more effective to learn AI agents that assist with the human’s strategy, which is an orthogonal paradigm that has rich prior literature.

---

> > ### Comment · Reviewer_RC1m · 2023-08-19
> >
> > I really appreciate the responses, which addressed some of my concerns. Having reviewed the rebuttal and other reviews, I agree that further experiments and real user studies would strengthen the work. At the same time, it's noteworthy as an initial endeavor to influence human behavior through offline RL with human-human interaction. I would maintain my score.

---

### Author Rebuttal · Authors · 2023-08-10

Based on reviewer feedback, we have performed multiple additional evaluations. We reference each of them in our individual responses to each reviewer, but also provide an overview of the new results below.

In Figure 1, we compute the reward improvements of our proposed method across all the experimental layouts. This improvement is computed as the difference between the reward accumulated in the last 100 timesteps, and the first 100. By showing a noticeably positive improvement, we aim to show that our method is successfully changing the behavior of its human partner via influence.

In Figure 2, we evaluate our proposed method (along with all baselines) against a scripted policy that will always perform the suboptimal high-level action, i.e. pick up onions when tomatoes yield higher reward. Here, we see that our proposed approach actually performs worse than baselines. This is to show that the reward improvement is actually due to influence, and not our learned agent solving the tasks more successfully independently.

In Figure 3, we visualize the learned latent strategies (as 2-dimensional embeddings obtained via PCA). Here, we show that different high-level actions map to different points in the space. This is to provide some interpretability on the learned latent strategies.

In Figure 4, we add an evaluation of an agent learned via self-play (specifically Fictitious Co-play). Because the self-play approaches often do not account for suboptimal partners, the approach performs much worse than ours in the generalization tasks where the human partner is guaranteed to be strategically suboptimal.

---

### Decision · Program_Chairs · 2023-09-21

**Decision:**

Accept (poster)

**Comment:**

This paper investigates how to leverage offline reinforcement learning on a dataset of human-human interactions to learn to influence human behavior toward desired outcomes. Human-subject studies are conducted to evaluate the proposed approach.

This is a borderline paper. All reviewers agree that the paper is addressing a timely and relevant question. The approach of using offline reinforcement learning in human-human interactions is novel and practical. The main debates are on whether the current set of experimental results sufficiently supports the claims, e.g., whether the approach indeed "influences" humans towards more optimal behavior. The authors have provided additional results to support the claims though there are differing opinions on whether more comprehensive experiments are needed.

Overall, considering the importance of the research question and the novelty of the approach, I tend to recommend accepting the paper. However, I'd recommend the authors take the reviewers' comments into account and provide more discussion on the implications and limitations of the results and how they relate to the claims of the paper.